# Zinc Supplementation Forms Influenced Zinc Absorption and Accumulation in Piglets

**DOI:** 10.3390/ani11010036

**Published:** 2020-12-27

**Authors:** Fen-Fen Liu, Md. Abul Kalam Azad, Zhi-He Li, Jing Li, Kai-Bin Mo, Heng-Jia Ni

**Affiliations:** 1Key Laboratory of Agro-Ecological Processes in Subtropical Region, Hunan Provincial Key Laboratory of Animal Nutritional Physiology and Metabolic Process, Hunan Research Center of Livestock and Poultry Sciences, South Central Experimental Station of Animal Nutrition and Feed Science in the Ministry of Agriculture, Institute of Subtropical Agriculture, Chinese Academy of Sciences, Changsha 410125, China; liufenfen0327@163.com (F.-F.L.); azadmak@isa.ac.cn (M.A.K.A.); 17347124127@189.cn (Z.-H.L.); lijing16099@163.com (J.L.); mokaibin2008@outlook.com (K.-B.M.); 2Department of Animal Science, Hunan Agriculture University, Changsha 410125, China; 3College of Veterinary Medicine, South China Agricultural University, Guangzhou 510642, China

**Keywords:** zinc, zinc methionine, metallothionein, mineral concentration, pig

## Abstract

**Simple Summary:**

High levels of inorganic Zn were supplemented to meet the nutritional requirements of Zn in piglets, which results in the development of intestinal microbial resistance and environmental pollution. Previous research indicated that organic Zn sources had higher bioavailability than inorganic Zn. There is limited information on the comparison of dietary Zn source dissolved in water with other Zn supplementation forms in piglets. Therefore, the present study was designed to compare the effects of three Zn supplementation forms (Zn-Met in the feed, ZnSO_4_ in the feed, and ZnSO_4_ dissolved in drinking water) on growth performance, Zn accumulation, activities of Zn-containing enzymes, expression of MT, and Zn transporters in piglets. In conclusion, Zn-Met and ZnSO_4_ dissolved in drinking water had higher bioavailability in piglets.

**Abstract:**

The study aimed at determining the effect of different zinc (Zn) supplementation forms on Zn accumulation, activities of Zn-containing enzymes, gene expression of metallothionein (MT), and Zn transporters in piglets. Eighteen piglets were randomly divided into three groups: (a) a basal diet supplemented with 150 mg/kg Zn from Zn methionine (Zn-Met) in the feed (Zn-Met group), (b) a basal diet supplemented with 150 mg/kg Zn from Zn sulfate (ZnSO_4_) in the feed (ZnSO_4_, _feed_ group), and (c) a basal diet supplemented with the same dose of Zn as in ZnSO4,_feed_ group but in water (ZnSO_4_, _water_ group). The results showed that Zn-Met added in feed and ZnSO_4_ dissolved in drinking water significantly improved (*p* < 0.05) the Zn concentration in liver and jejunum and the apparent digestibility of Zn in comparison with the ZnSO_4_ added in feed. In addition, dietary Zn supplementation as Zn-Met significantly increased (*p* < 0.05) the activity of alkaline phosphatase (AKP) in the jejunum of piglets in comparison with the ZnSO_4_, _feed_ group. Furthermore, the Zn-Met and ZnSO_4_, _water_ groups showed an improved total superoxide dismutase activity (T-SOD) in the ileum as compared to the ZnSO_4_, _feed_ group. Meanwhile, the qPCR and western blot results showed that Zn-Met and ZnSO_4_ dissolved in drinking water increased the expression of MT in the jejunum in comparison with the ZnSO_4_ added in the piglets’ feed. However, different Zn supplementation forms had no effect on the mRNA expressions of Zip4 and ZnT1 transporters. In conclusion, Zn-Met added in feed and ZnSO_4_ dissolved in drinking water had higher bioavailability in piglets.

## 1. Introduction

Zinc is one of the most essential trace minerals elements for animals, owing to its key role as an important component involved in forming numerous metalloenzymes and transcription factors [1]. Zn deficiency, usually caused by inadequate dietary Zn intake, affects growth, brain functions, and immune activity [2,3,4]. In pig production, the widely used forms of Zn are inorganic (ZnSO_4_, ZnO) because of their low cost and commercial preference. However, excessive feeding of high levels of inorganic Zn to piglets can stimulate resistance in the gut microbiota and may also result in a substantial excretion of Zn into the environment [5,6].

Research evidence has indicated that organic Zn sources is recommended widely in animals because of their higher bioavailability than inorganic Zn [7,8]. Xie et al. [9] reported that piglets with body weight of 11 kg fed diet supplemented with a lower dose of Zn-Met replacing 100 mg/kg ZnSO_4_ had no negative effects on nutrient digestibility, growth performance, and serum metabolites, and indeed, supplementing 50 mg ZnSO_4_ plus 25 mg Zn-Met to piglets exerted the beneficial effects of Zn digestibility and immune functions. Furthermore, Li et al. [10] found that in 38 weeks old laying hens dietary Zn-Met had beneficial effect on Zn accumulation in tissues, intestinal morphology, and the mRNA expression of metallothionein (MT) in the intestinal as compared to ZnSO_4_ supplementation in the diet. However, there is limited information on the comparison of dietary Zn source dissolved in water with other Zn supplementation forms in piglets. Therefore, based on these studies of foregoing, it was hypothesized that different Zn supplementation forms might influence effect growth performance, Zn accumulation, activities of Zn-containing enzymes, and the expression of MT and Zn transporters in piglets.

## 2. Materials and Methods

### 2.1. Animals, Diets and Experimental Design

The experimental design and procedures used in the present study were approved by the Animal Care and Use Committee of the Institute of Subtropical Agriculture, Chinese Academy of Sciences (IACUC # 201302).

Eighteen healthy male piglets (Duroc × (Danish Landrace x Yorkshire), 2nd and 3rd parities) with an average body weight of 14.47 ± 0.51 kg were randomly assigned to three dietary treatments with 6 animals each. The experiment lasts for 21 days. The treatment groups were follows: (a) Zn-Met group; a basal diet supplemented with 150 mg/kg Zn from Zn-Met in the feed, (b) ZnSO_4_, _feed_ group; a basal diet supplemented with 150 mg/kg Zn from ZnSO_4_ in the feed, and (c) ZnSO_4_, _water_ group; a basal diet supplemented with the same dose of ZnSO_4_ dissolved in water as ZnSO_4_, _feed_ group. The same dose Zn supplementation with three different patterns was based on limiting feed and water which containing ZnSO_4_. The drinking water method of piglets in ZnSO_4_, _water_ group was according the method described by Kaewtapee et al. [11]. Briefly, the loss of water in the home-made bottles which contains ZnSO_4_ was collected by a tray under the pen throughout the experimental period. The daily feed intake and ZnSO4 in drinking water intake are shown in Appendix A. The methionine content in the diets was maintained for all groups by adding methionine in the feed (i.e., for ZnSO_4_, _feed_ and ZnSO_4_, _water_ groups). Zinc sulfate (ZnSO_4_; Aladdin Biochemical Reagent Company, Shanghai, China) and Zn-Met (Tanke Group, Guangzhou, China) were pre-added in the premix or dissolved in drinking water to maintain their concentrations in the different dietary treatments. The composition nutrient levels of the basal diet met the National Research Council (2012) nutrient recommendation, as shown in Table 1.

### 2.2. Housing and Sample Collection

The piglets were housed in metabolic cages (1.5 m length × 0.8 m width) individually, and the room temperature was maintained at 25 °C. At the end of the experiment, 12 h after the last feeding, the piglets were weighed and sacrificed by electric shock (120 V, 200 Hz) for the sample collection. Liver and intestinal segments (the middle section of duodenum, the middle section of jejunum, the middle section of ileum) were collected. The intestine segments and liver were rinsed with physiological saline and quickly cut into lengths of 2 to 3 cm, then these segments were frozen in liquid nitrogen and stored at −80 °C for further analysis. One portion of the duodenum, jejunum and ileum, and the liver samples were cut into pieces of approximately 20 g per sample and were then stored at −20 °C for further Zn, Fe, and Cu analysis.

### 2.3. Chemical Composition of Feed Analysis

Laboratory analyses were carried out on feed samples using the standard AOAC (1995) procedures to determine digestible energy, crude protein, crude fat, and crude fiber content [12]. Lysine, methionine plus cysteine, and threonine were determined using the method of Miaomiao et al. [13].

### 2.4. Mineral Concentration Analysis

The mineral concentration of each sample was measured according to the method described by Cheng et al. [14]. Approximately 2 g of each fresh sample was weighed in triplicate and mixed with 10 mL of nitric acid and perchloric acid mixture (4:1, *v*/*v*). Then, the samples were kept for 12 h at room temperature and then digested to obtain clear digested solutions. The mixture was then heated to 80 °C for 1 h, followed by 120 °C for 1 h, 180 °C for 1 h, 220 °C for 1 h, and then maintained at 260 °C until dried to ash. The dried samples were suspended in 10 mL 1% nitric acid and filtered before analysis. The final solutions were analyzed for mineral concentrations using an Inductive Coupled Plasma Emission Spectrometer (ICP-720ES; Agilent, Palo Alto, CA, USA). The mineral concentrations were expressed as micrograms of mineral per gram of liver and intestinal segments.

### 2.5. Apparent Digestibility of Zn

The apparent digestibility of Zn was determined using the method of Lu et al. [15]. TiO_2_ was added to all diets at 5 g/kg as an external marker. At d18–d21, feces samples were harvested from each pigle, t and then the feces were dried at 65 °C for 24 h. The method of the concentrations of TiO_2_ and Zn in feces was described above. The apparent digestibility of Zn was calculated using the following formula.
Apparent digestibility of Zn (%) = 100 × [(Zn/TiO_2_)_feed+water_ − (Zn/TiO_2_)_faces_]/(Zn/TiO_2_)_feed+water_

### 2.6. Activity of Zinc Containing Enzymes

Zinc containing enzymes including total superoxide dismutase (T-SOD), alkaline phosphatase (AKP), and 5’-nucleotidase (5′-NT) were analyzed by using the commercially available kits (Nanjing Jiancheng Bioengineering Institute, Nanjing, China) according to the manufacturers’ instructions. T-SOD, AKP and 5′-NT were determined separately at 550 nm, 520 nm, and 680 nm with Multiscan Spectrum (Infinite M200 PRO, TECAN, CH).

### 2.7. Real-Time Quantitative PCR Analysis

The total RNA of liver, duodenum, jejunum, and ileum was extracted using Trizol reagent (Invitrogen, Carlsbad, CA, USA). The concentration of RNA was determined using a NanoDrop 2000 spectrophotometer (Thermo Scientific, Wilmington, DE, USA) and its integrity verified by electrophoresis on a 1% agarose gel. The cDNA was reverse transcribed from 1 μg of total RNA using the Revert Aid Reverse Transcriptase (Takara, Japan), then used for evaluating gene expression. The primers used for target genes are presented in Table 2. The qPCR was performed in a 10-μL reaction volume including 0.5 μM of each forward and reverse primer, 2 μL of cDNA, 2 μL of DEPC treated water, and 5 μL of SYBR Premix Ex Taq (Takara Bio Inc., Japan). The relative expression levels of genes were performed using the Lightcycler-480II system (Roche Diagnostics GmbH, Mannheim, Germany). The PCR cycling condition was 40 cycles at 94 °C for 40 s, 60 °C for 30 s and 72 °C for 35 s. Zn-Met group served as the control group and ZnSO_4_, _feed_ and ZnSO_4_, _water_ groups served as treatment. The relative expression was expressed using the formula 2^−(∆∆Ct)^, where ∆∆Ct = (Ct_Target_ − Ct_β-actin_)_treatment_ − (Ct_Target_ − Ct_β-actin_)_control_ [16,17]. Relative expression was normalized and expressed as a ratio to the expression in the Zn-Met group.

### 2.8. Western Blot Analysis

Western blot analysis was done according to the method described by Yin et al. [18]. Protein extracts were prepared in RIPA Lysis Buffer (Beyotime Biotechnology Inc., Shanghai, China). Approximately 30 μg proteins were subjected to SDS-PAGE electrophoresis. The duration of the electrophoresis was 30 V for 30 min, 80 V for 30 min, and 120 V for 30 min. Then, the proteins were transferred onto a PVDF membrane (Millipore, MA, USA) and blocked with 5% non-fat milk in Tris-Tween buffered saline for 1.5 h. Anti-Metallothionein rabbit polyclonal to antibody (ab233289, 1:5000, Abcam) and anti-beta actin mouse monoclonal antibody (60008-1, 1:5000, Proteintech) were used. The antibodies were incubated for 12 h at 4 °C. After incubation with the HRP-conjugated secondary antibodies (ZB-2301, 1:5000, ZSGB), the bands were visualized using the Alpha Imager 2200 software (Alpha Innotech Corporation, San Leandro, CA, USA).

### 2.9. Statistical Analysis

Data were analyzed by one-way analysis of variance followed by a Tukey’s honestly significant difference test using SPSS (version 22) [19]. *p* values of <0.05 were considered to indicate statistical significance among treatments.

## 3. Results

### 3.1. Effect of Zinc Supplementation Forms on Growth Performance and Apparent Digestibility of Zn

The results for piglets’ growth performance are presented in Table 3. The results show that there was no significant difference in growth performance between the groups with different Zn supplementation forms. The results showed that the ZnSO_4_, _feed_ group had lower (*p* < 0.05) apparent digestibility of Zn in comparison with the Zn-Met and ZnSO4, _water_ groups. The results also showed that the ZnSO_4_, feed group had lower (*p* < 0.05) apparent digestibility of Zn compared with the Zn-Met and ZnSO4, _water_ group.

### 3.2. Effect of Zinc Supplementation Forms on Tissue Mineral Concentrations

The results are summarized in Table 4. The ZnSO_4_, _feed_ group showed a lower (*p* < 0.05) concentration of Zn in the piglets’ liver and jejunum as compared to the Zn-Met and ZnSO_4_, _water_ groups. However, different Zn supplementation forms had no effect on Cu and Fe status in the piglets’ liver, jejunum, and ileum.

### 3.3. Effect of Zn Supplementation Forms on Zn-Containing Enzymes in Duodenum, Jejunum, Ileum and Liver

As presented in Table 5, the ZnSO_4_, _feed_ group had lower (*p* < 0.05) jejunal AKP activity in comparison with the Zn-Met group. Furthermore, the T-SOD activity in the ileum was lower (*p* < 0.05) in the ZnSO_4_, _feed_ group as compared to the Zn-Met and ZnSO_4_, _water_ groups.

### 3.4. Effect of Zn Supplementation Forms on MT mRNA and Protein Expressions in the Duodenum, Jejunum, Ileum and Liver

Figure 1 and Figure 2 showed that the mRNA expression and protein abundance of MT in the jejunum of the Zn-Met group and ZnSO_4_, _water_ group piglets were higher than those in piglets from ZnSO_4_, _feed_ group (*p* < 0.05). In addition, the ZnSO_4_, _feed_ group significantly decreased the expression of MT in the ileum in comparison with the Zn-Met group (*p* < 0.05). Meanwhile, supplementation of ZnSO4 in drinking water increased the MT mRNA expression and protein abundance in the liver as compared to the Zn-Met and ZnSO_4_, _feed_ groups (*p* < 0.05). However, there was no significant difference in MT mRNA expression and protein abundance in the duodenum between groups.

### 3.5. Effect of Zn Supplementation Forms on ZIP4 and ZnT1 mRNA Expressions in Duodenum and Jejunum

As presented in Figure 3, no significant difference was observed in ZIP4 and ZnT1 mRNA expressions in duodenum and jejunum between groups with different Zn supplementation forms.

## 4. Discussion

The present study showed that the different Zn supplementation forms had no effect on the growth performance of piglets. Xie et al. [9] also reported that there was no significant difference in the growth performance of pigs fed diets supplemented with different levels or sources of Zn. A recent study by Zhang et al. found that the piglets fed with 20, 40, and 80 mg Zn/kg as zinc amino acid complex had a similar growth performance to pigs fed with 40 mg Zn/kg as ZnSO_4_. Mallaki et al. [20] observed a higher average daily gain and lower feed conversion ratio in male lambs supplemented with Zn-peptide as compared to those fed ZnSO_4_. Therefore, the addition of 150 mg/kg of Zn in piglets’ diets might be an adequate level for the growth of pigs from 15 kg to 24 kg body weight. Another possible explanation might the shorter intervention time of the Zn supplementation and the smaller sample size of the experiment. However, growth performance parameters may not be an ideal index to evaluate Zn requirements, tissue Zn concentration and Zn-dependent enzyme activities are considered as sensitive criteria to determine the requirements.

Although Zn supplementation form had no effect on the growth performance of piglets, there is still a question whether it can influence the bioavailability of Zn in pigs or not. Phytates are identified as the limiting factor of Zn absorption in monogastrics by forming insoluble phytate–Zn complexes. The concentration of feed phytate is approximately 0.8~2.5 g/kg [21]. Zn-Met, organic metal chelates composed of Zn chelated with Met in coordinate covalent bonds, chelate is stable in the small intestine, may minimize the formation of Zn-phytate complex and allow more Zn to be absorbed by the epithelial cells in the small intestine [22]. In contrast, inorganic form, ZnSO_4_, is easily dissociated in the stomach and intestine, therefore, Zn-phytate complex is formed, rendering a lower absorption of Zn [23]. The current study demonstrated that Zn-Met added in feed improved Zn apparent digestibility. This is in agreement with previous studies, in which organic Zn sources such as Zn-Met, Zn-proteinate and Zn-glycinate had a greater bioavailability as compared to inorganic Zn forms [23,24,25]. Moreover, this study found that ZnSO_4_ dissolved in drinking water also had higher Zn apparent digestibility in comparison with ZnSO_4_ added in feed. That might be because piglets did not drink ZnSO_4_ dissolved in water at feeding time, which reduced the contact of Zn^2+^ with phytic acid in the feed.

Zinc from different sources was found to be digested and absorbed in the intestine, where intestinal MT captured a small part of the absorbed Zn, and the remaining major part is captured by hepatic MT [26]. Thus, intestinal and liver Zn contents can accurately reflect the differences in Zn absorption and utilization from different Zn sources. In the present study, the results showed that supplementation of Zn-Met added in feed and ZnSO_4_ dissolved in drinking water significantly enhanced Zn contents in the liver and jejunum of piglets. This is in agreement with a previous study by Liu et al. [27], in which the supplementation of organic Zn to growing-finishing pigs significantly increased the concentrations of Zn in the liver in comparison with the inorganic Zn supplementation. The results indicated that ZnSO_4_ dissolved in drinking water was better absorbed and utilized than ZnSO4 added in feed. Meanwhile, Zn supplementation forms had no effect on Cu and Fe contents in the liver, jejunum, and ileum. Therefore, the results suggest that Zn-Met added in feed and ZnSO_4_ dissolved in drinking water improved Zn absorption and had no effect on Cu and Fe absorption.

Zn is an important ingredient in zinc-containing metalloenzymes, including T-SOD, AKP, and 5′-NT. Previous studies showed that T-SOD, AKP, and 5′-NT activities were significantly affected by Zn deposition and can be used for the evaluation of body Zn status [28]. Several studies have shown that T-SOD, AKP, and 5′-NT levels were affected by different Zn supplementation forms [29,30]. Our current study showed that Zn-Met added in feed and ZnSO_4_ dissolved in drinking water could increase the activities of AKP in the jejunum and T-SOD in the ileum as compared to the ZnSO_4_ added in feed. Li et al. [30] found that pigs fed a basal diet supplemented with 40, 60, 80, and 100 mg/kg of Zn as Zn-Met increased the AKP activity in serum as compared to the pigs who were fed a basal diet supplemented with 80 mg of Zn/kg as ZnSO_4_. Thus, the present study results supported the hypothesis that the increase of AKP and T-SOD activities might result from higher Zn availability in the body. Moreover, Zn-Met added in feed and ZnSO_4_ dissolved in drinking water might supply more available Zn for enzymes to maintain their normal activity.

The increased Zn concentrations and Zn-containing enzymes activities in the jejunum and liver resulting from the supplementation of Zn-Met added in feed and ZnSO_4_ dissolved in drinking water that were observed in present study may be associated with Zn absorption and transportation [31]. MT and zinc transporters (ZnT and ZIP families) are the proteins involved in intracellular zinc homeostasis through influxing, chelating, sequestrating, coordinating, releasing, and effluxing Zn. MT is a group of low-molecular-weight metal-binding proteins that bind zinc with high affinity and to serve as an intracellular zinc reservoir [32]. MT expression is induced by zinc elevation, and thus, zinc homeostasis is maintained [26,32]. The small intestine is the primary site for Zn absorption. Hence, the increasing amounts of MT in the small intestine indicate that Zn was more available for absorption by the animal. The present study showed that Zn-Met added in feed and ZnSO_4_ dissolved in drinking water could increase the mRNA expression of MT and protein abundance in jejunum and ileum in comparison with the ZnSO_4_ added in fed. In a previous study, the addition of different Zn sources has been found to increase the mRNA expressions of MT in intestinal porcine epithelial cells [33]. Moreover, Huang et al. also found that the increased levels of dietary Zn-glycine supplementation significantly enhanced the expression of MT1 mRNA in the duodenum [34], which may suggest that Zn-Met added in feed and ZnSO_4_ dissolved in drinking water improved the Zn availability in comparison with ZnSO_4_ added in the fed. Another possible reason might be explained as Zn-Met added in feed and ZnSO_4_ dissolved in drinking water could stimulate Zn absorption capacity compared to the ZnSO_4_ added in fed. The hepatic MT bound to majority of Zn in the liver, MT concentrations in the liver was found positively associated with the Zn concentration in pigs [35].

ZIP and ZnT transporter family proteins which transport Zn^2+^ across biological membranes play crucial role in maintaining Zn homeostasis [36]. ZIP4 is essential for Zn accumulation from the gut lumen, locates at the apical membrane of enterocytes. The level of ZIP4 mRNA expression in the intestine can be upregulated under Zn deficiency and downregulated under increased Zn concentration [37]. In contrast, intestinal ZnT1 transport cytosolic Zn^2+^ into the circulation [38]. Previous study also has shown that different Zn supplementation form affect Zn absorption, when dietary Zn increases, absorption of Zn decreases [39]. The present study found that different that Zn supplementation form had no effect on the mRNA expressions of ZIP4 and ZnT1 in the duodenum and jejunum of pigs. It may be resulted from different Zn supplement patterns contain the same does of Zn that did not have much effect on the transcription of ZIP4 and ZnT1.

## 5. Conclusions

In summary, Zn-Met added in feed and ZnSO_4_ dissolved in drinking water was more effective for Zn accumulation in the jejunum and for enzyme activities in the jejunum and ileum. These results are also supported by the mRNA expression and protein abundance of MT in jejunum and ileum. However, different Zn supplement patterns did not have much effect on the transcription of ZIP4 and ZnT1 in the duodenum and jejunum of pigs. The concrete mechanism of Zn-Met added in feed and ZnSO_4_ dissolved in drinking water needs to be further studied.

## Figures and Tables

**Figure 1 animals-11-00036-f001:**
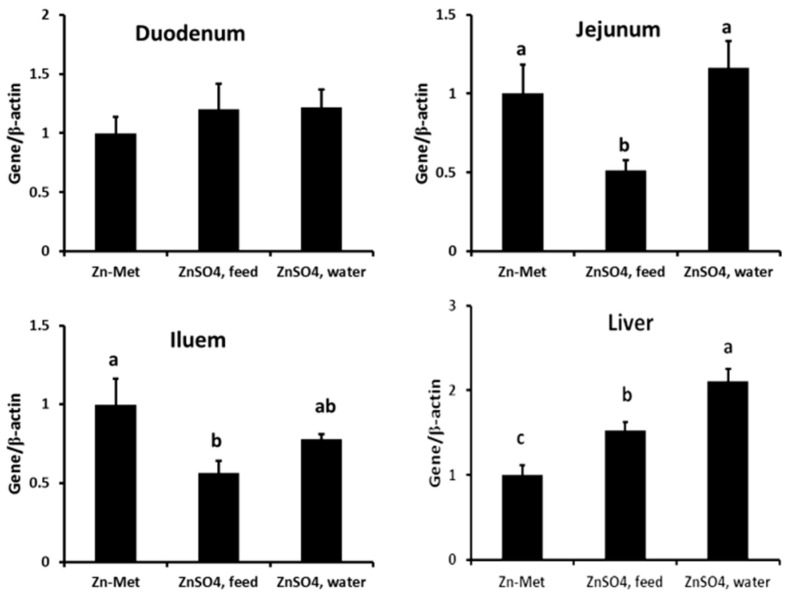
Effect of different Zn supplementation forms on MT mRNA expression in the duodenum, jejunum, ileum, and liver of the piglets. a–c: different letters were significantly different among groups (*p* < 0.05).

**Figure 2 animals-11-00036-f002:**
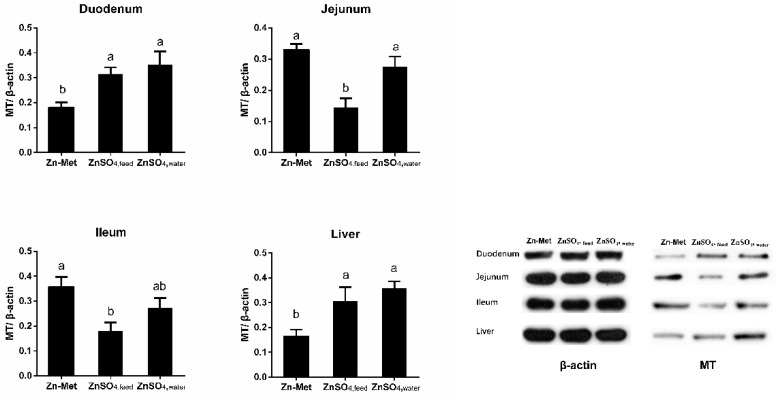
Effect of different Zn supplementation forms on MT protein levels in the duodenum, jejunum, ileum, and liver of the piglets. a, b: different letters were significantly different among groups (*p* < 0.05).

**Figure 3 animals-11-00036-f003:**
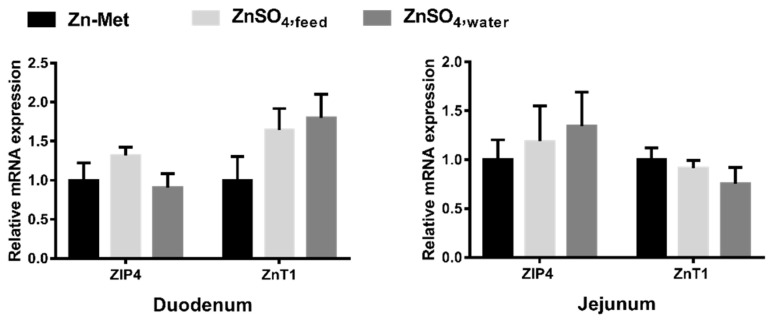
Effect of different Zn supplementation forms on ZIP4 and ZnT1 mRNA expressions in the duodenum and jejunum of piglets.

**Table 1 animals-11-00036-t001:** Composition of basal diet (as-fed basis) for piglets.

Ingredient	Content (%)	Analyzed Value	
Corn	63.80	Digestible energy (MJ.kg^−1^)	14.05
Soybean meal	19.80	Crude Protein (%)	18.27
Whey powder	4.30	Crude fat (%)	4.20
Fish meal	9.00	Crude fiber (%)	2.55
Soybean oil	0.80	Lysine (%)	1.37
Lysine	0.38	Methionine + Cysteine (%)	0.76
Threonine	0.09	Threonine (%)	0.88
Tryptophan	0.01	Calcium (%)	0.80
Limestone	0.52	Phosphorus (%)	0.64
Salt	0.30	Zinc (mg/kg)	70.46
1% premix ^1^	1.00	Copper(mg/kg)	47.47
Total	100.00	Iron(mg/kg)	195.09

^1^ Premix supplied per kilogram of diet: vitamin A_1_, 11 809.4 IU; vitamin D_3_, 5480 IU; vitamin E, 43.84 IU; vitamin K_3_, 10.96 mg; vitamin B_2_, 13.70 mg; vitamin B_1_, 5.48 mg; vitamin B_6_, 6.58 mg; vitamin B_12_, 0.05 mg; vitamin C, 109.60 mg; choline 1644 mg; salt, 2959 mg; sugar, 7891 mg; niacin/niacinamide, 54.80 mg; folic acid, 2.74 mg; biotin, 0.33 mg; lysine, 5477 mg; methionine, 1754 mg; threonine, 2192 mg; Cu as CuSO_4_·5H_2_O, 38.22 mg; Fe as ferrous bisglycinate chelate, 112 mg; I as KI, 0.19 mg; Mn as MnSO_4_·H_2_O, 71.25 mg; Se as Na_2_SeO_3_, 0.27 mg; Co as CoCl_2_, 0.11 mg.

**Table 2 animals-11-00036-t002:** Primers used in this study.

Gene	Accession No.	5′-3′ Primer Sequence
β-actin	XM_003357928	F: CGTTGGCTGGTTGAGAATC
		R: CGGCAAGACAGAAATGACAA
ZIP4	XM_021090449	F: TGCTGAACTTGGCATCTGGG
		R: CGCCACGTAGAGAAAGAGGC
ZnT1	NM_001139470	F: CCAGGGGAGCAGGGAACCGA
		R: TCAGCCCGTTGGAGTTGCTGC
MT	NM_001001266.2	F: CTGTGCCTGAAGTCTGGGGAA
		R: CACAGAAAAAGGGATGTAGCATG

F, forward; R, Reverse; ZIP4, solute carrier family 39 member 4; ZnT1, zinc transporter 1; MT metallothionein.

**Table 3 animals-11-00036-t003:** Effects of zinc supplementation forms on growth performance and apparent digestibility of Zn in piglets.

Items	Zn-Met	ZnSO_4_,_feed_	ZnSO_4_,_water_	*p*-Value
Average daily gain (kg)	0.443 ± 0.0102	0.471 ± 0.0531	0.435 ± 0.0462	0.209
Average daily feed intake (kg)	0.730	0.730	0.730	1.000
Final body weight (kg)	23.28 ± 0.263	23.90 ± 0.432	23.15 ± 0.312	0.274
Feed: gain ratio	1.67 ± 0.0403	1.56 ± 0.0732	1.69 ± 0.0714	0.265
Apparent digestibility of Zn (%)	40.29 ± 5.15 ^a^	23.43 ± 4.14 ^b^	46.21 ± 2.62 ^a^	0.004

^a,b^ Values with different letters were significantly different (*p* < 0.05).

**Table 4 animals-11-00036-t004:** Effect of different Zn supplementation forms on mineral concentrations in the jejunum, ileum, and liver of the piglets.

Tissue	Items	Zn-Met	ZnSO_4_,_feed_	ZnSO_4_,_water_	*p*-Value
Liver	Zn (mg/kg)	248.00 ± 44.7 ^a^	162.50 ± 40.00 ^b^	233.50 ± 20.4 ^a^	0.002
	Cu (mg/kg)	16.72 ± 2.64	14.54 ± 2.19	13.48 ± 3.08	0.148
	Fe (mg/kg)	111.10 ± 20.8	82.30 ± 46.20	87.90 ± 21.0	0.279
Jejunum	Zn (mg/kg)	19.19 ± 0.49 ^a^	16.95 ± 0.43 ^b^	19.76 ± 0.75 ^a^	0.008
	Cu (mg/kg)	1.86 ± 0.21	1.64 ± 0.13	1.74 ± 0.08	0.571
	Fe (mg/kg)	55.06 ± 5.52	56.15 ± 6.41	38.78 ± 2.81	0.991
Ileum	Zn (mg/kg)	15.07 ± 0.54	17.03 ± 0.57	16.87 ± 0.73	0.075
	Cu (mg/kg)	1.33 ± 0.26	1.32 ± 0.20	0.90 ± 0.23	0.336
	Fe (mg/kg)	8.40 ± 0.70	8.84 ± 0.95	7.79 ± 0.39	0.595

Means in the same row with different letters were significantly different among groups (*p* < 0.05).

**Table 5 animals-11-00036-t005:** Effect of different Zn supplementation forms on Zn-containing enzymes in the duodenum, jejunum, ileum, and liver of the piglets.

Tissue	Item	Zn-Met	ZnSO_4_,_feed_	ZnSO_4_,_water_	*p*-Value
Duodenum	T-SOD	52.16 ± 4.74	58.00 ± 7.77	55.37 ± 4.60	0.785
	AKP	15.22 ± 1.92	15.63 ± 2.31	19.49 ± 1.66	0.273
	5′-NT	10.85 ± 1.19	8.19 ± 0.75	9.47 ± 1.06	0.216
Jejunum	T-SOD	72.67 ± 4.81	64.89 ± 3.80	82.82 ± 7.96	0.124
	AKP	39.17 ± 2.64 ^a^	29.85 ± 1.60 ^b^	36.35 ± 30.06 ^ab^	0.031
	5′-NT	15.81 ± 1.38	12.24 ± 1.03	17.71 ± 2.51	0.116
Ileum	T-SOD	68.73 ± 4.1 ^a^	54.26 ± 2.38 ^b^	66.11 ± 4.62 ^a^	0.039
	AKP	34.85 ± 1.70	31.02 ± 1.89	36.13 ± 3.95	0.406
	5′-NT	15.95 ± 1.40	16.36 ± 1.67	15.11 ± 1.08	0.815
liver	T-SOD	176.34 ± 72.06	232.20 ± 32.12	195.37 ± 32.34	0.478
	AKP	32.20 ± 12.00	37.64 ± 10.78	16.48 ± 5.78	0.317
	5′-NT	34.49 ± 7.95	45.97 ± 5.57	45.68 ± 8.79	0.503

Means in the same row with different letters were significantly different among groups (*p* < 0.05).

## Data Availability

All data used in the current study are available from the corresponding author on reasonable request.

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
