# Peer review of "Zinc Supplementation Forms Influenced Zinc Absorption and Accumulation in Piglets"

_animals, 2020, doi:10.3390/ani11010036_

Round 1
Reviewer 1 Report
Comments are in the attachment.

Reviewer 2 Report
The manuscript entitled “Zinc supplementation patterns influenced Zinc absorption and accumulation in piglets” has been improved, however, native English review is strongly recommended.
There are a few points, mainly minor issues that have to be further clarified:
- I am not confident with the term “Zinc supplementation pattern”. Pattern is a rhythm or a configuration, and since there is no information on the distribution of Zn uptake, the title is not proper. I suggest using “form of Zinc supplementation”.
- line 41: make it more generous 11 kg or a range (10-15 kg) rather than 10.69 kg, also in line 207: 14-23 kg.
- the restricted feed and water consumption should be mentioned in section of “Animals, diets and experimental methods” rather than in section “Housing and sample collection”.
- Please, check the units in Table 1. It is doubtful whether the Ca and P is reported in mg/kg.
- Instead of stone powder, I suggest limestone or CaCO3
- In some cases, the reference used is apparently strange: why reference 13 is used for dietary Lys, Met, Cys, and Thr? The paper (Safwat et al, 13) is about determination of amino acids in muscle fibre and not in feed. In line 105: the apparent digestibility method is referring to a rabbit study. There are numbers of piglet study describing the method of determination the digestibility, therefore it should be specified why a rabbit study in more relevant in the present study.
- Still the calculation routine of determination of apparent digestibility in water dissolved Zn treatment is not clear. I suggest the following formula: ((Zn/TiO2)feed+water - (Zn/TiO2)feces) / (Zn/TiO2)feed+water, if it is right.
- In the discussion section (line 216-221), please, give an assumption on the phytate level of the feed (even if it was not measured)
- The new sentence in line 230-231 suits better to one sentence further (after the sentence starts as “This is in agreement with previous study by Li et al. (27)…”
Reviewer 3 Report
The majority of my comments with regard to the manuscript have been addressed accordingly. The revised version of the manuscript has significantly improved.
Reviewer 4 Report
Dear authors,
Thank you for your interesting work. I have ranked your work into minor revisions.
L20 “than that in comparison” Language
L56: Please provide more information on the family structure (no. of sire and dams)
L91: Format error: italic?
L156, L170, L222, L233 … in comparison with … or as compared with … or … against ...
Fig. 1 and Fig. 2: Link within the text is missing.
L148: “Data are presented as … “. No need to mention in this part.
L159: At least 3 significant digits should be used in the result table. This is violated by the items “… daily gain” and “… daily feed intake”
L189: Formatting error: Font size
L193: Fig. 3 instead of Fig. 2
Author Response
Please see the attachment.

This manuscript is a resubmission of an earlier submission. The following is a list of the peer review reports and author responses from that submission.
Round 1
Reviewer 1 Report
All comments are included in the attached file.

Reviewer 2 Report
The manuscript reports an experiment studying the effect of form of Zn supplementation on the digestibility and bioavailability of Zn.
My main problem is that although the feed intake was limited thus identical daily rations were fed, but there is no indication what was the water intake. One of the treatments was dissolved Zn supplementation, therefore it is crucial to know what was the total Zn intake in different treatment groups. Actually, not the Zn concentration (g/kg) but the Zn supply in g/d makes the effect. It is not clear how the digestibility coefficient was calculated in the treatment with water dissolved Zn.
Without reporting the water intake and the total Zn supply the results on digestibility are not convincing. The discussion is based on the statement that the higher digestibility resulted a better bioavailability.
Minor issues:
In the Introduction the authors state that the most widely used source of Zn is the inorganic for. Therefore it is interesting why the organic Zn is considered as control in the study.
Why stone powder was used?
Dietary Zn should be reported in Table 1.
More detail is needed which particular segments (where from, how long) of gut were sampled.
I suppose the trial was 3 weeks long, but it is speculated only based on sentence in L82.
Is it correct that the se was that high in Jejunal AKP in treatment 3? Was there any outlayer?
Reviewer 3 Report
Attached please find my comments and recommendations for this manuscript. Thank you.

Reviewer 4 Report
Dear authors,
Thank you for your interesting work. I have ranked your work into mayor revisions.
In general, I have a problem with the use of abbreviations, which should be described when they were used first.
L21: Explanation of AKP, explained in L 100
L22: T-SOD, explained in L 100
L46: MT, explained in the abstract.
...
L35: What do you mean by commercial preference.
L41: „low level of Zn-Met“. I assume, that there is a more exact description available.
L56: Duroc x (Danish Landrace x Yorkshire) => better to identify sire and dam line
L56: Please provide more information on the family structure (no. of sire and dams)
L46: The role of the MT-gene is in my opinion not clear. Please provide more information.
L118: In your statistical analysis piglet birth weight is missing. Why or do you think, that this effect does not have an impact on your dependent variables?
L129: No. of digits of the SE should be increased.
L135: More information about the distribution and within group standard deviations of the phenotype would be useful.
L153: Format errors
L100: What are the treatment and control groups? Need more information
L169: Description of figure 1 is incomplete. Labels of the columns (western blot) are missing.
L169: There is no interpretation or explanation of the protein levels (western blot results). Why?
L169: May be I‘ m wrong, but your statistical test should allow to test the mt-RNA differences control group (Zn-Met, standardized to 1) against treatment groups. Taking that into account, it is not allowed to present the significance levels between the treatment groups?
Figure 1 and 2 are different in resolution and quality.
L191: Typo „was might“
